# Asymmetric Neutral Point Diode Clamped Topology with Reduced Component Count for Switched Reluctance Machine Drive

**Pieter Antonie Scholtz** *[ID] **and Michael Njoroge Gitau**

Faculty of Engineering, Built Environment and Information Technology, University of Pretoria, Hatfield, Pretoria 0002, South Africa; njoroge.gitau@up.ac.za
* Correspondence: pieterantonie@live.com

**Abstract:** The Reduced Asymmetric Neutral Point Clamped converter topology for unipolar driven, multiphase switched reluctance machines is proposed in this paper. This topology shares similarities with the conventional NPC and Asymmetric-NPC topologies, however it is unique in that the components for the capacitor string and outer semiconductor switches are shared among all the phases for a reduced component count. Some switching state combinations are not possible during commutation overlap between motor phases, resulting in minor torque transients during regenerative braking. A custom modulation scheme is implemented with fixed frequency, phase-shifted carrier waveforms that allow for automatic balancing of the neutral point voltage and interleaved switching of the semiconductor switches. A simple torque observer control architecture is used with minor adjustments for arbitrating torque contribution priorities between phases during handover.

**Keywords:** Pulse Width Modulation (PWM); switched reluctance motor (SRM); Neutral Point Clamped (NPC) converter

## 1. Introduction

The unipolar nature of a dc-excited switched reluctance machine (SRM) can allow for semiconductor reductions in standard drive topologies. Recent publications on the use of multilevel converters for SRMs, focus on leveraging this unipolar nature of the machine and the selective phase commutation for the design of further reduced switch topologies.

For SRM multilevel drive strategies, boosting voltage at the beginning and end of a stroke is a common strategy used to reduce the rise times of current during excitation and demagnetisation. This reduces the required commutation overlap and extends the operable speed range. Due to the short duration of use, there is an incentive for sharing the boost circuitry between phases.

A variety of front-end topologies like the Passive Boost Converter [1], Integrated Multiport Converters with boost capacitor [2] and Quasi-Z-Source topology [3], Modular Multilevel converter with decentralised batteries [4] and the dual output Cuk converter with input side power factor correction [5] are available in the literature and regulate the intermediate dc bus voltage levels based on speed. However, as noted by [6], these topologies fall short in that they do not allow derating of the semiconductor switches that are directly driving the motor, as these still need to withstand the full bus voltage from the supply.

Asymmetric variants of traditional multilevel topologies like the five level Asymmetric Neutral Point Clamped (ANPC), Asymmetric Flying Capacitor (AFC) and Asymmetric Cascaded Cell Half Bridge (ACCHB) were initially proposed in [7] and nine level ANPC and Asymmetric Modular Multilevel (AMM) by [8]. The component reductions considered by [7] were not optimal for the ANPC and AFC topologies, and was later improved on by [6,9] respectively. These topologies allow for derating semiconductor switches directly

driving the motor and have all the traditional benefits of multilevel topologies such as reduced filtering requirements. Relative to the front-end converters, the multilevel circuitry and voltage boosting has to be duplicated per motor phase which has certain cost and volume implications.

Component reductions evaluated by [6] for the ANPC converter, included sharing the capacitor string between two converter arms and removing some of the redundant clamping diodes on the top right arm and bottom left leg compared to [7]. The consideration of additional switch reductions based on selective phase commutation during multiphase operation and sharing of switches between phases is a trend in SRM converter literature. No reduced topologies with switch sharing between phases are currently found for the AFC and ACCHB topologies while variations of the ANPC topology [6] based on these considerations can be found in the literature. In these reduced topologies, switch components are shared between multiple phases during operation.

In [10,11], modified ANPC topologies (relative to [6]) are proposed without the bottom right leg switch and clamping diode and sharing the uppermost switch among all phases. The dc supply is also connected over the bottom capacitor. This requires the upper capacitor to be charged up before operation as a boost voltage source but reduces the total capacitance requirement. Ref. [12] shares the top switch between phases and uses full bridge modules for additional error tolerance and lower conduction losses. The converter has a wye configuration for the phases, an extra converter arm for the neutral point and can be realised using off-shelf six-pack modules and a single phase NPC module. Ref. [13] proposed a reduced ANPC topology sharing the upper left arm switches and the bottom-most, right leg switch with wye connection of the phases. In contrast to the topology proposed by [6], the wye connection of the phases in these converters compromise on the operational range of the converter, as simultaneous excitation and demagnetisation at the maximum voltage is not possible due to the shared current path.

Due to excluded bottom clamping diode and switch, current can only be regenerated into the maximum bus voltage of [10,11]. This limits the controllability of current ripple during regenerative braking and reduces efficiency. The topologies in [10,12] can be driven from a lower voltage source, but do not have access to the maximum voltage on start-up.

The topologies proposed in [10,11] do not allow for the use of multiple voltage levels during generation and [12,13] have prominent torque harmonics due to the wye configuration of phases. Ref. [12] is not a true asymmetric configuration which would significantly increase the cost of the drive. Some of the component reductions are promising and there is scope to investigate their impact and identify further reductions that will not inhibit converter performance in four quadrants of operation.

A need for research on natural, carrier modulation schemes was identified for the asymmetric multilevel topologies in [14] where switching losses and total harmonic distortion of the input current should become predictable and controlled. Topologies like [8] and [6] use PWM control with lookup tables for switching state selection. In [6], the carrier modulation is additionally used to coordinate sampling of the phase current and switching frequency. The other ANPC variants of [10–13] use multilevel hysteresis schemes with arbitrary state selection from lookup tables and delta modulation by sampling the output of a hysteresis controller. Carrier modulation schemes were proposed in [14] for the ANPC [6], AFC [9] and ACCHB [7] topologies. However, a method for extending the modulation to be applicable for further switch reductions has not been previously covered and is addressed in this study.

This paper proposes a new Reduced-ANPC (RANPC) topology as an alternative to the ANPC topology from [6] and reduced component variants of [10–13,15]. The proposed topology achieves similar functionality for driving a three-phase SRM from a dc bus in a unipolar fashion relative to [6] but offers further component reductions due to sharing of components between phases. The RANPC shares the uppermost and bottommost switch of the ANPC topology between all the phases, reducing the required components without compromising on the multilevel operational capability for motor or generator mode

operation. The natural carrier modulation strategy from [14] is adapted to accommodate the additional component reductions. A simple torque observer control architecture is used the regulate the torque with minor alterations for improving the torque handover between phases during commutation.

The remainder of the paper is structured as follows: The basic topology with the shared components and modulation method will be discussed in Section 2. This will also examine the effect of component sharing during simultaneous conduction which can compromise the performance of some of the switching states. Simultaneous conduction typically happens during torque handover at the commutation overlap between phases. Proposed control is discussed in Section 3, Simulink simulation setup and machine model in Section 4 and then results in Section 5.

## 2. Switch-Sharing and Introducing RANPC Topology

The proposed topology is shown in Figure 1. Component sharing is achieved through parallelisation of the inner AHB sections inside the ANPC topology whilst sharing clamping diodes, capacitor string and the outer switches T1 and T4 between all the phases. The switching states for a single phase are described in Table 1 and are the same as for the ANPC converter [6].

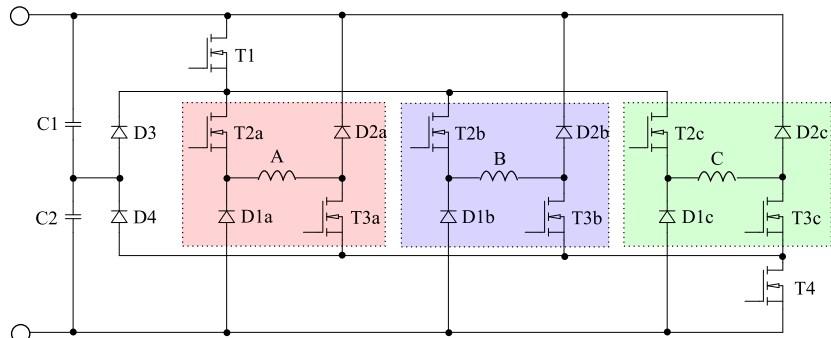

**Figure 1.** Proposed three phase Reduced ANPC (RANPC) converter. Note the sharing of the clamping diodes, outer switches, and the parallelisation of the inner AHB sections. These are highlighted for each phase.

**Table 1.** Switching states for a single phase of the RANPC converter.

| State | T1 | T2 | T3 | T4 | Applied Voltage | Conduction Loss | Energy Transfer * | Condition |
|-------|----|----|----|----|-----------------|-----------------|-------------------|-----------|
| 1 | 1 | 1 | 1 | 1 | $V_{DC}$ | $-4V_{ce,sat}$ | | |
| 2 | 1 | 1 | 1 | 0 | $V_{C1}$ | $-3V_{ce,sat} - V_{fwd}$ | C2 ↑ C1 ↓ | |
| 3 | 0 | 1 | 1 | 1 | $V_{C2}$ | $-3V_{ce,sat} - V_{fwd}$ | C1 ↑ C2 ↓ | |
| 4 | 1 | 1 | 0 | 0 | 0 | $-2V_{ce,sat} - V_{fwd}$ | | $i_L > 0$ |
| 5 | 0 | 1 | 1 | 0 | 0 | $-2V_{ce,sat} - 2V_{fwd}$ | | $i_L > 0$ |
| 6 | 0 | 0 | 1 | 1 | 0 | $-2V_{ce,sat} - V_{fwd}$ | | $i_L > 0$ |
| 7 | 0 | 1 | 0 | 0 | $-V_{C1}$ | $-V_{ce,sat} - 2V_{fwd}$ | C2 ↓ C1 ↑ | $i_L > 0$ |
| 8 | 0 | 0 | 1 | 0 | $-V_{C2}$ | $-V_{ce,sat} - 2V_{fwd}$ | C1 ↓ C2 ↑ | $i_L > 0$ |
| 9 | 0 | 0 | 0 | 0 | $-V_{DC}$ | $-2V_{fwd}$ | | $i_L > 0$ |

* Shows the transfer of energy between sources. ↑ indicates charging (current flowing into source) and ↓ indicates discharging (current flowing from source).

Due to the sharing of the top and bottom switch (T1 and T4), some of the intermediate voltage levels are not possible during simultaneous conduction. This result in a higher output voltage for a specific switching state. These compromised switching states only occur during handover (overlap) between two phases. All the possible switching state combinations and their resulting output voltages are shown in Table 2 for a generalised

phase N and N + 1. The combinations that result in a compromised output voltage are highlighted.

**Table 2.** RANPC Phase voltage for phase N during simultaneous switching with Phase N + 1.

| N+1 \ N | State 5, 7, 8, 9 | State 2, 4 | State 3, 6 | State 1 |
|---|---|---|---|---|
| State 1 | $V_{DC}$ | $V_{DC}$ | $V_{DC}$ | $V_{DC}$ |
| State 2 | $V_{C1}$ | $V_{C1}$ | $V_{DC}$ | $V_{DC}$ |
| State 3 | $V_{C2}$ | $V_{DC}$ | $V_{C2}$ | $V_{DC}$ |
| State 4 | 0 | 0 | 0 | 0 |
| State 5 | 0 | $V_{C1}$ | $V_{C2}$ | $V_{DC}$ |
| State 6 | 0 | 0 | 0 | 0 |
| State 7 | $-V_{C1}$ | 0 | $-V_{C1}$ | 0 |
| State 8 | $-V_{C2}$ | $-V_{C2}$ | 0 | 0 |
| State 9 | $-V_{DC}$ | $-V_{DC}$ | $-V_{DC}$ | $-V_{DC}$ |

This table shows the switching states of phase N on the row headings, phase N + 1 on the column headings and phase voltage for N in the data section. Compromised output states, where the phase voltage for N deviates from the single phase model in Table 1 due to simultaneous conduction with N + 1, are highlighted.

Only two carrier signals are required for modulation of four switches across the entire operational range of the drive as shown in Figure 2a,b. These carriers are phase-shifted and can be shared among the modulation for individual phases. The interleaving increases the ripple frequency over the phase to be at twice the switching frequency of the individual switches.

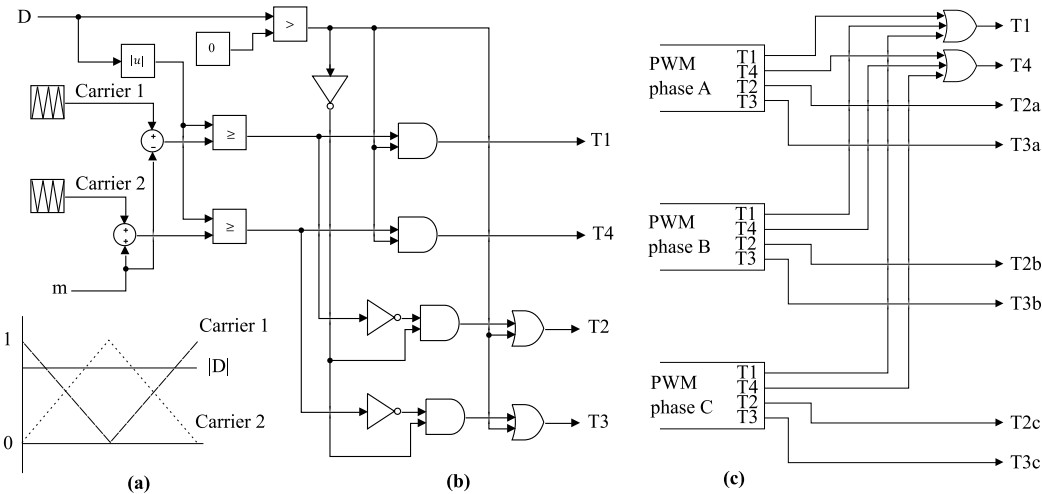

**Figure 2.** Phase-shifted carrier modulation scheme for the RANPC converter. Where (**a**) represents carrier waveforms and duty ratio as a graph, (**b**) represents the implementation of a single phase with a logic diagram. Capacitor balancing of the neutral point is implemented using control input *m*. (**c**) demonstrates the extension of single-phase modulation (in (**b**)) to multiple phases by logical OR operation of the shared switch gate signals (T1 and T4).

For excitation and motor mode operation, switching states 1–3 and 5 from Table 1 are utilised. The positive voltage is controlled through actuation of the switches T1 and T4 while T2 and T3 remain permanently on. For demagnetisation and generator mode operation, the negative voltage states are applied through actuation of T2 and T3 using states 5 and 7–9 while T1 and T4 remain permanently off if excitation has taken place and the phase current ($i_L$) is larger than zero.

Switching states 4 and 6 for zero voltage are considered redundant states and are not implemented in the proposed modulation. The single-phase modulation scheme from [14] is expanded to the multiphase RANPC converter by implementing a logic OR operation on the T1 and T4 gate signals as shown in Figure 2c. The combined switching states that are possible with the proposed component sharing and modulation are summarised in Table 2 for two adjacent phases N and N + 1. Phase N can be fully demagnetised using state 9 without sacrificing functionality of phase N + 1. However, operation of phase N to other operational states is compromised due to the component sharing. Operation of these compromised states are compensated for in control, but results in larger current ripple during generator mode handover periods.

An example of the gate signal combinations as implemented in the control of this drive during handover is shown in Figure 3 for (a) generator mode operation and (b) motor mode operation. The current path for these implemented states are shown in Figure 4.

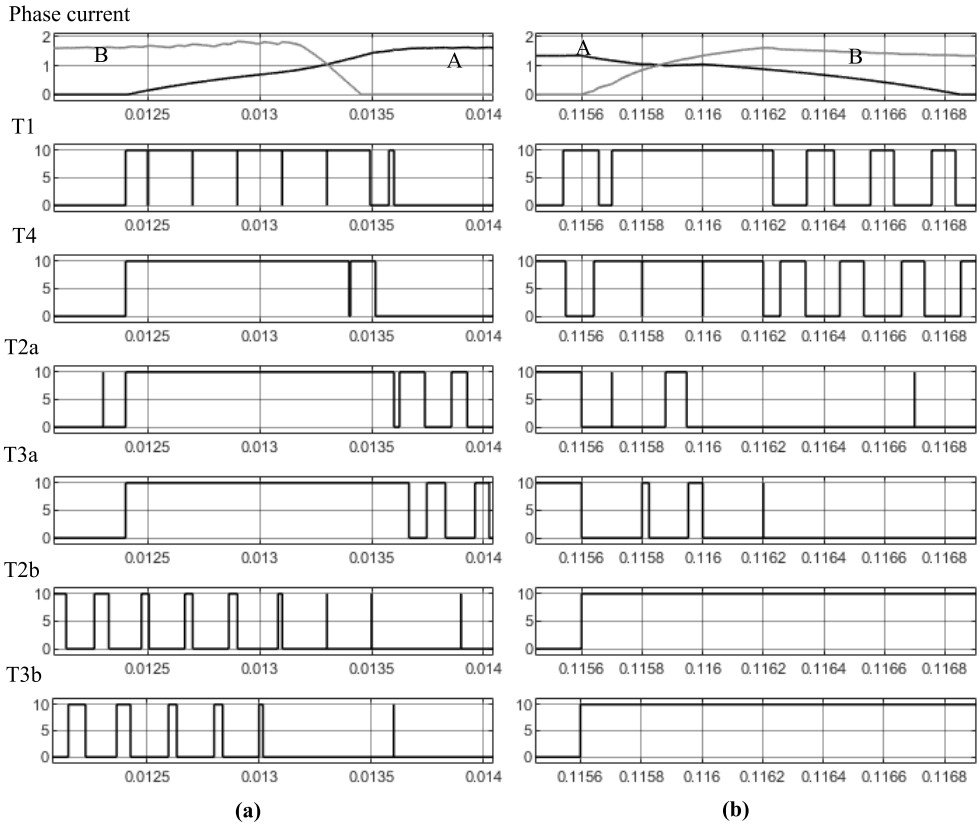

**Figure 3.** Gate signals during handover from (**a**) phase B to A for generator mode operation and (**b**) from phases A to B for motor mode operation.

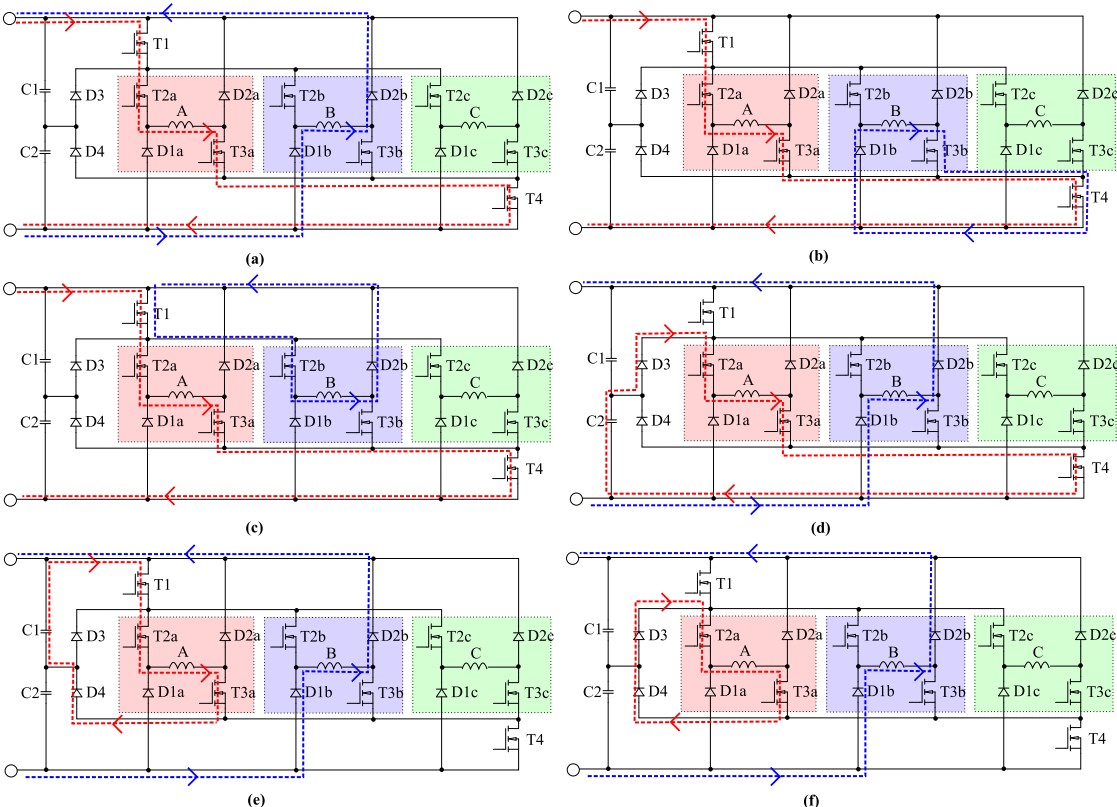

**Figure 4.** Equivalent circuits for simultaneous conduction of two phases. Where (**a**–**c**) shows phase A in state 1 and phase B in states 9, 6 and 4 respectively. Likewise (**d**–**f**) show phase B in state 9 and phase A in states 3, 2 and 5 respectively. The flow of current for the two phases during these states are indicated using the dotted lines.

*Sizing of Power Capacitor for RANPC*

For the RANPC the power capacitor string should be able to supply sufficient energy to maintain the rated current without a significant reduction in voltage. Both $C1$ and $C2$ should be able to supply half of the rated phase current ($I_{rated}$) for half of the switching period ($T_{sw}$) while maintaining the desired voltage ripple ($V_{C,ripple}$). This results in,

$$C1 = C2 = \frac{I_{rated}T_{sw}}{4V_{C,ripple}} \tag{1}$$

For the designed converter, $I_{rated} = 2$ A, $T_{sw} = 0.0002$ s, $V_{C,ripple} = (0.1)V_C$ where $V_C$ is the individual capacitor voltage at 50 V. This results in $C1 = C2 = 20$ µF.

## 3. Control

Due to the switch sharing, it is not recommended to use standard Torque Sharing Functions (TSFs), because some of the intermediate voltage levels are unavailable during simultaneous conduction. A torque observer control architecture based on the standard hysteresis DITC used by [12,16–18] is used in this study. It requires only a single lookup operation in the loop. The hysteresis controller was replaced with the proposed PWM controller from [14] adapted for multiphase RANPC, Figure 2.

The PWM controller allows for the implementation of the interleaved switching scheme with predictable input current harmonics, switching losses and maintenance of the neutral point voltage. Because the torque is non-linearly related to the phase current, this leads to a reduced dynamic performance. This could be improved by converting the torque reference to a flux linkage reference and using a flux linkage observer and control loop for

a linearised response [19]. For the static load used in this simulation, the proposed direct torque control, tuned with a PID compensator, adequately tracks the reference.

A priority selector block is added to select between feeding back the total torque versus the phase torque. This is required for generator mode operation because excitation occurs at the aligned position and is therefore slower than demagnetisation. By using the prioritisation scheme, it is possible to excite the incoming phase while the preceding phase compensates for the torque shortfall. This allows for a smooth torque handover. An additional control block for regulating the neutral point voltage with modulation input $m$ is also included. The control signal, $\theta_{ctrl}$, is used to delineate the turn-on and turn-off for a specific phase. The prioritisation signal $\theta_{prio}$ is only needed during generating mode and is generated as the logical AND of the preceding and current $\theta_{ctrl}$ signals. The phase control and phase prioritisation signals are shown in Figure 5 for motor mode operation and Figure 6 for generator mode operation. The final control scheme for the specific topology is shown in Figure 7.

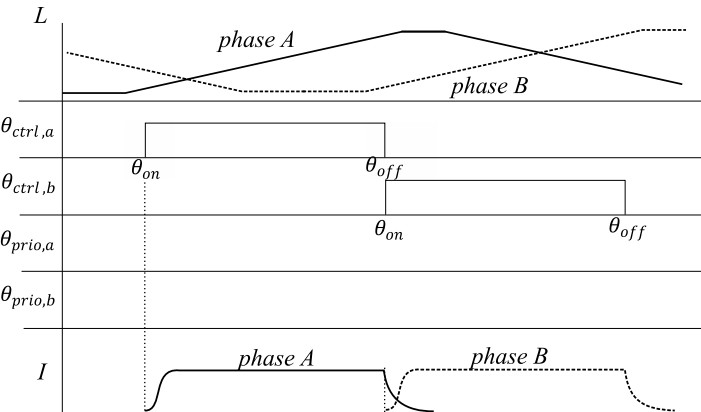

**Figure 5.** Control signals for Figure 7 illustrating the use of the $\theta_{ctrl}$ and $\theta_{prio}$ signals for two adjacent phases during motor operation.

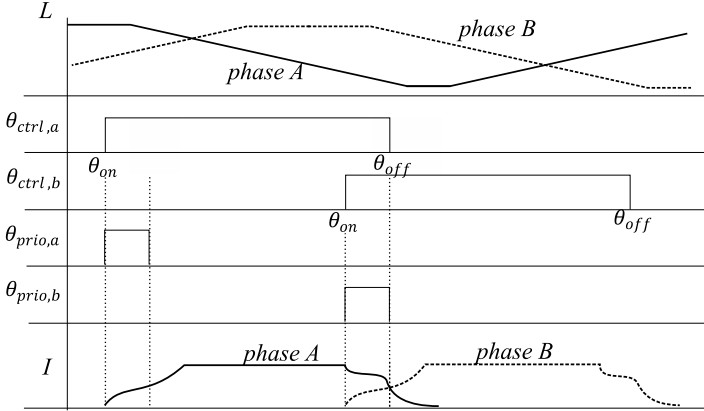

**Figure 6.** Control signals for Figure 7 illustrating the use of the $\theta_{ctrl}$ and $\theta_{prio}$ signals for two adjacent phases during generator operation.

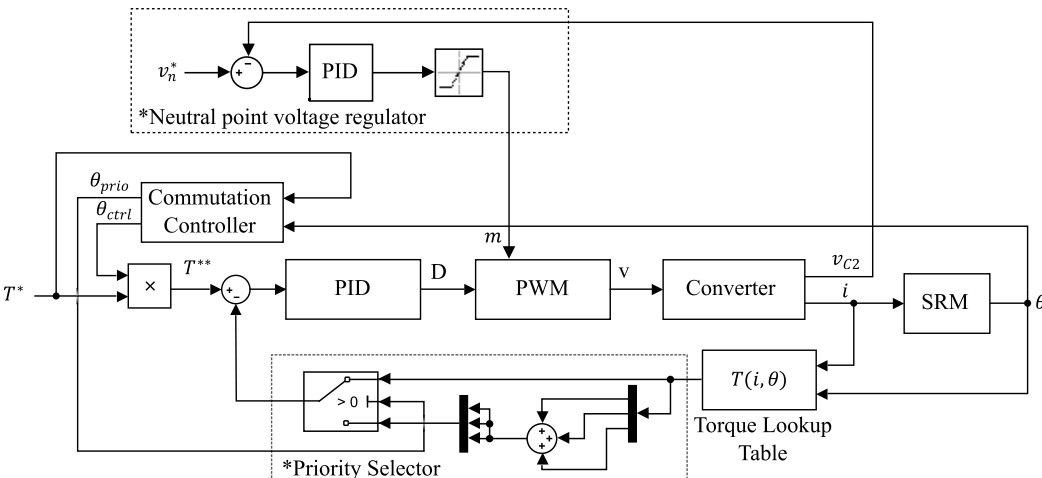

**Figure 7.** Torque control strategy with PID controller and PWM block. The priority selector block is introduced for smooth torque during handover of both motoring and generating operation. The neutral point voltage regulator controls the carrier waveform displacement relative to one another using control input *m* from Figure 2 to control the neutral point voltage to track the reference.

*Regulation of Neutral Point Voltage*

Figure 2 provides a control input *m* which can be used to level shift the carrier signals relative to one another. By manipulating this control input, the ratio between time spent discharging and charging the neutral point of the converter over a switching period can be varied. This is used to regulate the voltage balance between capacitors *C1* and *C2*. The effect of this control action is illustrated in Figure 8, where the small offsets are introduced between the carrier signals to extend the time spent either charging or discharging the neutral point.

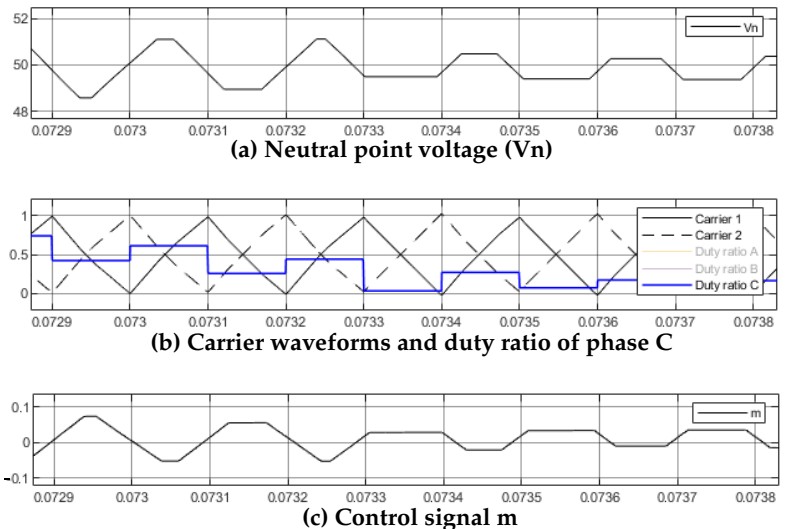

**Figure 8.** Illustrates the use of control input *m* and the modulation scheme of Figure 2a to manipulate the offset between carrier waveforms during motor mode operation. By manipulating *m*, it is possible to change the ratio between charging and discharging of the neutral point voltage of the converter, thereby regulating the neutral point voltage.

This control is implemented in the neutral point voltage regulator block of Figure 7. The output of the PID controller can optionally be constrained with a saturation block to limit the maximum corrective action that can be taken during a single switching period. The optimal selection of the saturation thresholds is not investigated in this paper but will depend on the maximum allowed voltage ripple of the neutral point voltage.

## 4. Materials and Methods

The proposed converter was modelled in Simulink. The machine parameters for an SRM with an unsaturated, linear induction profile are given in Table 3 and sectioned drawing of the motor is provided in Figure 9 as definition for the parameters. The salient poles on the rotor and stator of the motor are rectangular without any eccentric features (bevels, cut-outs, or chips). The transition from aligned to unaligned therefore shares the same profile as proposed in the models by [20,21] and is shown in Figure 10. The shape is approximated using a Fourier series as proposed by [22].

**Table 3.** Machine specifications.

| Model | | H55PWBKM-1850 |
|---|---|---|
| **Specification** | **Value** | **Description** |
| $R_s$ | 2.5 Ohm | Phase resistance |
| $L_a$ | 52 mH | Aligned inductance |
| $L_u$ | 9 mH | Unaligned inductance |
| $I_{rated}$ | 2.5 A | Rated current |
| $V_{rated}$ | 120 V | Rated voltage |
| $q$ | 3 | Phases |
| $P_s/P_r$ | 12/8 | Pole configuration |
| $Rg$ | 41.45 mm | Outer radii of the rotor |
| $Rsp$ | 41.75 mm | Inner radii of stator poles |
| $pws$ | 11.7 mm | Stator pole width |
| $pwr$ | 10.8 mm | Rotor pole width |
| $\beta_r$ | 14.971° | Rotor pole angle |
| $\beta_s$ | 16.110° | Stator pole angle |
| $K$ | 0.165 H/rad | Derivative of inductance relative to angular position $(L_a - L_u)/\beta_r$ |

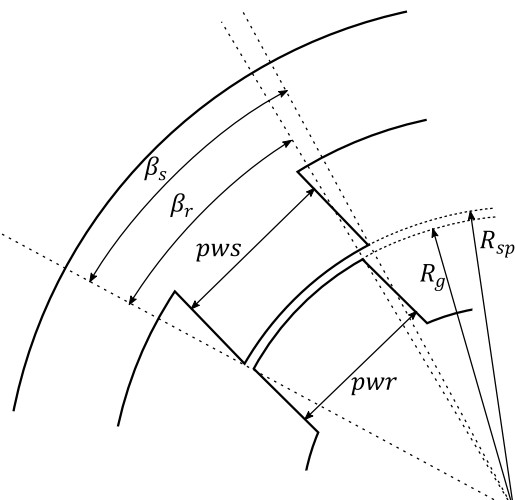

**Figure 9.** Definition of motor dimensions.

For simulation, this model used the start of mechanical overlap versus the actual magnetic overlap. There is an unknown offset angle [21] between the actual and predicted torque capability areas of the machine. This unknown offset of the real inductance profile is superimposed as a dotted line in Figure 10 and can be adjusted manually during practical implementation.

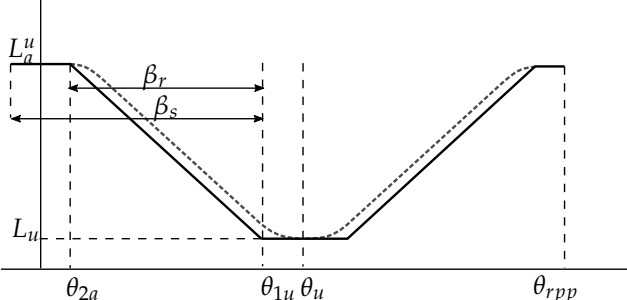

**Figure 10.** Unsaturated inductance profile for a standard SRM, Where $\beta_s$ is larger than $\beta_r$.

The simulation was set up across the operable speed range of the machine. The simulation starts by regeneratively braking against a high, rated speed in the opposite direction of rotation. The machine slows down due to the braking torque and eventually reverses the direction of rotation. At this point, the drive transitions into motoring mode operation and the machine starts to speed up again in the direction of the applied motoring torque.

## 5. Simulation Results

The resulting torque waveforms over the simulation period are presented in Figure 11. The torque is regulated to the set point however it is clear that there are some dips and transient spikes during the handover between phases. The regulated neutral point voltage over the same period presented in Figure 12, is seen to successfully regulate around the 50V set point during operation.

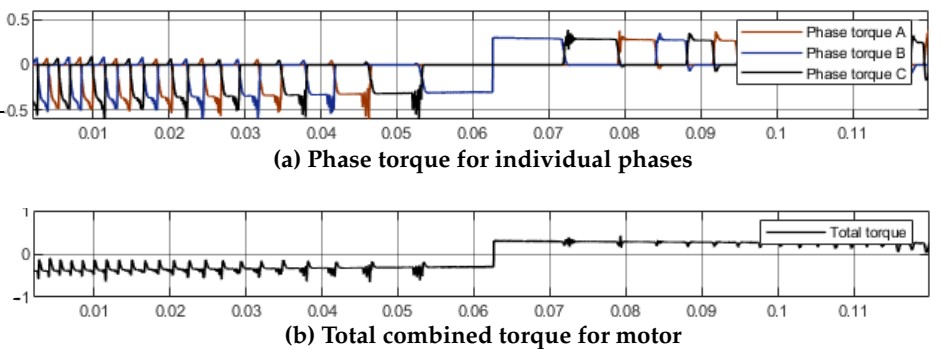

**Figure 11.** Tested torque range of the drive. Starts in generating mode with regenerative (torque regulated braking). Eventually reverses direction at time t = 0.062, transitioning into motor mode operation. The torque transients during generation are caused by regulation with the compromised states during handover.

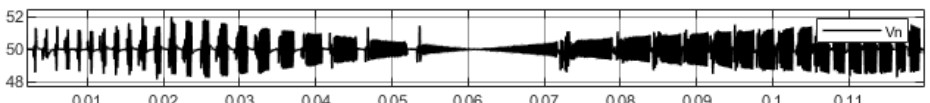

**Figure 12.** Regulation of neutral point voltage.

The gate signals synthesized by the control during handover between phases are shown in Figure 3 for generator mode operation in (a) and motor mode operation in (b). These figures illustrate the successful modulation of the duty ratio for overlapping, successive phases, and the specific gate signals for driving.

The input current, phase current, duty ratio and phase voltage waveforms are shown in Figure 13 for motor mode operation, and Figure 14 for generator mode operation.

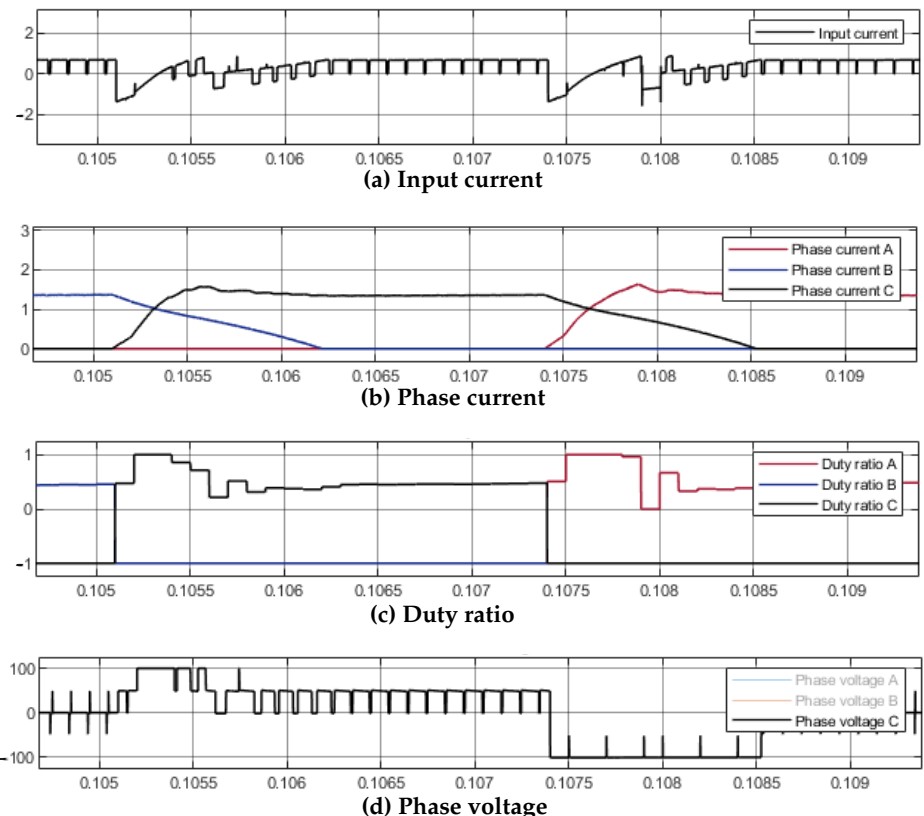

**Figure 13.** RANPC waveforms during motoring mode operation.

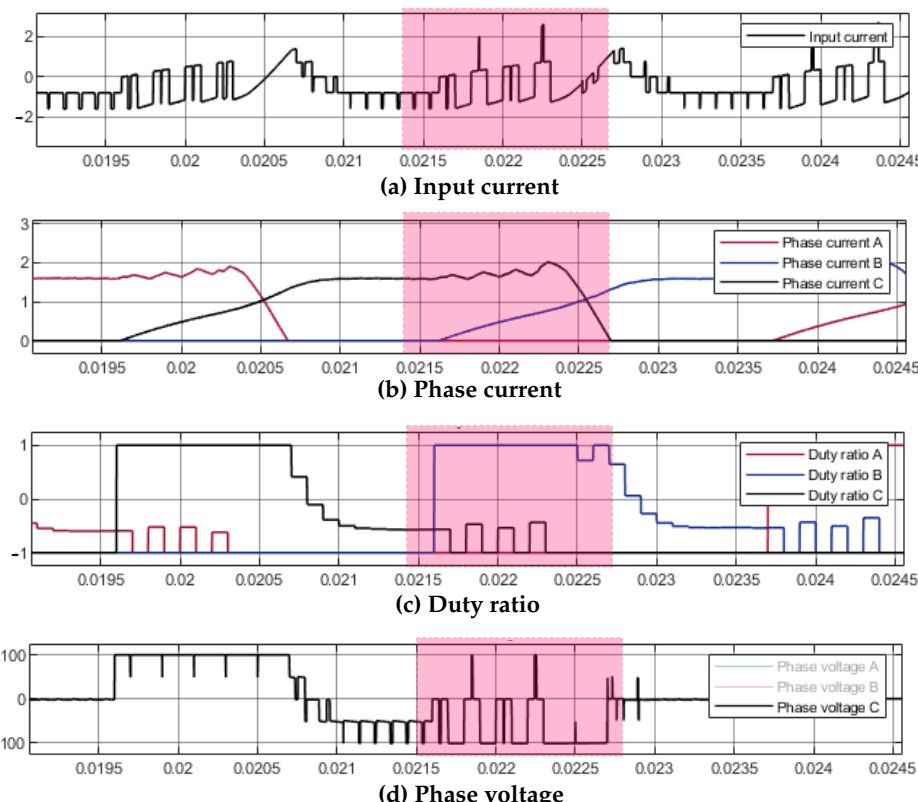

**Figure 14.** RANPC waveforms during generator mode operation. The effect of the compromised states are highlighted in red with phase current ripple introduced for phase C during overlap.

## 6. Discussion

The dips in total torque of Figure 11 during handover of torque at high speeds is inherent to the motor geometry used, where there is only a small overlap between the torque capability of phases for the three phase machine. This will improve when the overlap region is larger.

The transient spikes in the total torque waveform are due to the modulation with the compromised states and the resulting deviation of the phase current. This effect on phase current regulation is seen during generator mode operation when the preceding phase hands over to the incoming phase and the current ripple increase due to a loss of some intermediate voltage levels. This is illustrated in Figure 3a. The excitation of phase A forces T1, T4, T2a and T3a to be held high, implementing states 1–3 from Table 1. Phase B is regulating switch inputs T2b and T3b to implement states 5 and 7–9. However, because T1 and T4 are shared between the two phases, the controller is implementing states of a higher voltage level. This results in an oscillation of the duty ratio as shown in the highlighted regions of Figure 14 as the controller now regulates between a higher voltage and zero resulting in a period of increased input and phase current ripple.

During the motor mode operation shown in Figure 13, this effect is not present, as the demagnetisation is slow relative to the excitation during handover. This means that intermediate voltage levels are not required during excitation, and the compromised states do not affect torque performance.

## 7. Conclusions

The RANPC topology was proposed in this paper with a phase-shifted modulation scheme. A simple torque observer control structure, with minor adaptations for prioritising the torque contribution between phases, was used to demonstrate the functionality of the proposed topology. This topology incorporates switch sharing for a unipolar driven SRM to reduce the component count. The proposed modulation scheme avoids the arbitrary look-up operations for switching state selection in [6] and allows for interleaved switch operation with balancing of the neutral point voltage as part of the natural modulation. The topology utilises intermediate voltage levels during regenerative braking which is not possible in the existing literature of [10,11,13,15]. This capability is only constrained during handover of one phase to another, where the compromised states due to switch sharing cannot synthesise some of the intermediate voltage levels. The ripple current amplitude becomes larger for both input and phase current during this handover period which can also results in some high frequency torque transients. The impact of this higher current ripple will depend on the specific use case.

The emphasis of this work was on showcasing the functionality of the reduced component controller and the design steps taken with custom modulation and minor torque controller changes. Future publications on this topology could aim to characterise the impact on the stability and modeling of the torque control loop during handover between phases as well as the improvements in input current harmonic distortion which can now be described in a more predictable manner due to the use of carrier modulation.

**Author Contributions:** Conceptualization, P.A.S.; methodology, P.A.S.; software, P.A.S.; validation, P.A.S. and M.N.G.; formal analysis, P.A.S.; investigation, P.A.S.; resources, M.N.G.; data curation, P.A.S.; writing—original draft preparation, P.A.S.; writing—review and editing, P.A.S. and M.N.G.; visualization, P.A.S. and M.N.G.; supervision, M.N.G.; project administration, M.N.G. All authors have read and agreed to the published version of the manuscript.

**Funding:** This research received no external funding.

**Data Availability Statement:** Simulation files and results data are available on request.

**Conflicts of Interest:** The authors declare no conflict of interest. The funders had no role in the design of the study; in the collection, analyses, or interpretation of data; in the writing of the manuscript, or in the decision to publish the results.

## Abbreviations

The following abbreviations are used in this manuscript:

| | |
|---|---|
| SRM | Switched Reluctance Machine |
| ANPC | Asymmetric Neutral Point Clamped |
| AFC | Asymmetric Flying Capacitor |
| ACCHB | Asymmetric Cascaded Cell Half Bridge |
| AMM | Asymmetric Modular Multilevel |
| PWM | Pulse Width Modulation |
| RANPC | Reduced Asymmetric Neutral Point Clamped |

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
