# Peer review of "Asymmetric Neutral Point Diode Clamped Topology with Reduced Component Count for Switched Reluctance Machine Drive"

_energies, doi:10.3390/en15072468_

Round 1
Reviewer 1 Report
Thank you for selecting this topic. Kindly check the following comments to enhance your article significantly before publication.
- The abstract must be rewritten with concise technical information about this work.
- Please avoid the lumped references like 2-6, 7-11, etc. I have noticed that the references are not cited coherent manner inside the reference section. Authors should reframe the content in order to showcase the reference citation in a sequential manner.
- Also, the current version of the introduction lacks the technical literature survey. Please highlight the author's contribution at the end of the introduction section.
- Review felt that the topology is not a new one. Please justify with your answers.
- Interleaved carrier modulation is the same as phase shift modulation? the reviewer felt a little confused based on Fig. 2a. Kindly clarify it.
- Fig. 4 also creates little confusion, better add the arrow mark in the dotted line so that the readers will get clarity.
- What is the value of ripple taken for equation 1 in your work?
- How has the PID controller value been calculated?
- Fig.7 is also a simple closed-loop method for SRM motor drive. Please justify with your novelty.
Author Response
[1] The abstract must be rewritten with concise technical information about this work.
Abstract has been rewritten to clearly delineate the scope of application as well as emphasise what is being contributed
[2] Please avoid the lumped references like 2-6, 7-11, etc. I have noticed that the references are not cited coherent manner inside the reference section. Authors should reframe the content in order to showcase the reference citation in a sequential manner.
The lumped references have been removed in some cases and expanded in others to have more context. In the references section these are currently listed in order of appearance in the paper. There is however repetition of the use of some of the references (especially in the introduction section) Where different topologies are being compared against each other. The authors could unfortunately not find a way to rephrase these paragraphs as handling one reference at a time would become very verbose and would make it harder to follow the comparison.
[3] Also, the current version of the introduction lacks the technical literature survey. Please highlight the author's contribution at the end of the introduction section.
The technical literature survey was present in the previous version of the introduction. In this revision, the authors have tried to improve coherency and flow so that this would be more obvious.
Please refer to line numbers 91 - 98 in the newest revision
[4] Review felt that the topology is not a new one. Please justify with your answers.
The closest topology in terms of functionality is currently found in Reference 6 "An Asymmetric Three-Level Neutral Point Diode Clamped Converter for Switched Reluctance Motor". This specific topology is also a variation of the NPC converter with certain changes for asymmetric operation. It did not consider any of the additional component reductions proposed in this paper like the sharing of semiconductors between phases. The proposed topology has a significantly lower component count.
Compared to other reduced ANPC topologies of references [10]-[13] There are significant differences in dealing with the sharing of either the top and bottom switch as well as how the dc supply is connected to the capacitor string. There are also larger implications in that many of the shared switch topologies that cannot excite and demagnetise simultaneously at the maximum voltage available nor necessarily use the full spectrum of intermediate voltages in all four quadrants of operation.
[5] Interleaved carrier modulation is the same as phase shift modulation? the reviewer felt a little confused based on Fig. 2a. Kindly clarify it.
The modulation makes use of two phase-shifted carrier waveforms that result in interleaved switching of the semiconductor switches at a frequency that is lower than the current ripple frequency at the output. Occurrences of the use of "interleaved" to describe the modulation scheme has now been changed to phase-shifted to avoid any ambiguity.
[6] Fig. 4 also creates little confusion, better add the arrow mark in the dotted line so that the readers will get clarity.
Arrows have been added to Fig. 4 to indicate the direction of current flow.
[7] What is the value of ripple taken for equation 1 in your work?
Parameters for equation 1 has been added, peak to peak Voltage ripple is taken as maximum 5V at rated current of 2A.
[8] How has the PID controller value been calculated?
The PID controller values were tuned by hand in the simulation test bench. The optimal torque control was considered to be outside the scope of this work and the results for torque performance from the simulation was included as a proof of concept to show that the sharing of switches do not hamper the converter's ability to synthesise a smooth torque profile.
%The scope of the contribution was delineated to just the development of the topology and its successful modulation.
%The winding back electromotive force (emf) and electrical time constant vary significantly within one electrical cycle of the SRM due to the change in the inductance. Also the torque is non-linearly related to the current through the windings. Initially the small signal model was derived based on the method put forth by "A Study of Current Controllers and Development of a Novel Current Controller for High Performance SRM Drives" by H.K. Bae and R. Krishnan, however adapting the model for the consideration of torque proved to be quite complicated and eventually the calculated tuning was replaced with a hand tuned PID controller as this sped
[x] Fig.7 is also a simple closed-loop method for SRM motor drive. Please justify with your novelty.
%The simple closed loop was modified with the priority selector block for regulating smooth torque. This is different from the usual commutation controller as it uses a prioritisation signal to either feed back the sum of the total torque experienced by the machine or the individual torque contributions of a single phase based on which phase has been prioritised. This allows the prioritised phase to contribute the maximum torque while the secondary (non-prioritised) phases contribute the torque shortfall for a smoother handover. One can refer to Fig. 5 and Fig. 6 for specific examples of what these control signals look like.
%It should be emphasised that this topology does not work well with torque sharing functions (which is a common approach used with SRM drives) because of the limited access to the intermediate voltage levels during handover between phases.
% The emphasis on the torque controller architecture has been reduced in the conclusion and it is made clear that a simple torque observer has been used with minor adaptations for prioritising torque during handover between phases.
% Future work paragraph has been added recommending an investigation into modeling and optimally compensating for the torque control loop especially during handover between phases.
%
Reviewer 2 Report
The paper demonstrates a new Asymmetric Neutral Point Diode Clamped converter with reduced Component Count for Switched Reluctance Machine Drive. The authors properly presented modulation technique based on interleaved carrier modulation and the control method of the torque control strategy with the PID controller algorithm to improve. Simulation and experimental results are used to validate the effectiveness of the new method well. The method looks promising for industrial design and applications. Only two minor comments for your considerations are the followings:
- Most figures of simulations and experiments are in high definition, but figures 3 and 8 are too small. Specially the fonts are not clear and sharp enough to present the multiple results in the same figures and coordinate systems.
2. The author describes modulation technique and control methods, but the equations are not presented. It should be improved because there is only one equation in all papers.
Author Response
[1] Most figures of simulations and experiments are in high definition, but figures 3 and 8 are too small. Specially the fonts are not clear and sharp enough to present the multiple results in the same figures and coordinate systems.
Figures 3 and 8 have been resized so the font size is similar to that of the other figures and more readable.
[2] The author describes modulation technique and control methods, but the equations are not presented. It should be improved because there is only one equation in all papers
Derivation of the small signal model and optimisation of the torque controller was not considered to be part of the scope of this article which is why small signal equations and controller equations have not been included. The PID compensation was tuned by hand.
Regarding the equations for modulation, this has been represented with a logic diagram in Fig. 2. Could the reviewer perhaps elaborate on what additional equations he/she would like to see in the article?
Round 2
Reviewer 1 Report
Thank you for your response letter.